# Effectiveness of the mushroom technique versus morcellation in en bloc bipolar prostate enucleation for prostates over 80 mL

Zoltán Kiss[ID][1]*, Mihály Murányi[ID][1], Attila Nagy[2], Tibor Flaskó[1]

**1** Department of Urology, University of Debrecen, Debrecen, Hungary, **2** Department of Health Informatics, Faculty of Health Sciences, University of Debrecen, Debrecen, Hungary

* kissjz85@gmail.com

## Abstract

### Background and objectives

Anatomical endoscopic enucleation of the prostate (AEEP) has revolutionized the surgical treatment of benign prostatic hyperplasia, offering advantages over transurethral prostate resection (TURP). This study aimed to compare the effectiveness of the mushroom technique versus morcellation in en bloc transurethral bipolar prostate enucleation (TUEB) for prostates > 80 mL, hypothesizing that morcellation would reduce operative time.

### Patients and methods

We retrospectively reviewed 234 TUEB procedures performed between January 2018 and March 2024 at a tertiary university hospital. Patients were divided into two groups: the mushroom technique group (n = 116) and the morcellation group (n = 118). Demographic and clinical data were collected and outcomes, such as operative time, efficacy, and complications, were analyzed.

### Results

The median operative time was significantly longer in the mushroom cohort (80 [60–90] min) than in the morcellation cohort (60 [50–70] min) (p < 0.001). Efficacy, which was measured in g/min, was higher in the morcellation group (1.25 [1.01–1.44] g/min) than in the mushroom group (0.9 [0.76–1.03] g/min) (p < 0.001). Linear regression analysis showed that the prostate volume significantly influenced the operative time of the mushroom technique more than that of the morcellation technique. Complications were similar across the groups, with no need for blood transfusion or conversion to TURP.

**Data availability statement:** The anonymized data supporting the findings of this study are available as a supplementary file accompanying this manuscript.

**Funding:** The author(s) received no specific funding for this work.

**Competing interests:** The authors have declared that no competing interests exist.

**Abbreviation:** AEEP, anatomical endoscopic enucleation of the prostate; ASA, American Society of Anesthesiologists; BMI, body mass index; BPH, benign prostatic hyperplasia; IIEF-5, International Index of Erectile Function; IPSS, International Prostate Symptom score;P-SA, prostate specific antigen; PVR, post-void residual; Qave, average flow rate;Qmax, maximum flow rate; QoL, quality of life; TSUI, transient stress urinary incontinence; TUEB, transurethral bipolar prostate enucleation; TURP, transurethral resection of the prostate.

## Conclusions

Morcellation is an efficient method for tissue removal during bipolar prostate enucleation particularly for larger prostates, owing to its shorter operative time and consistent efficacy. Although the mushroom technique is a viable alternative, it is less efficient for larger glands. Nevertheless, both techniques demonstrated similar safety profiles and functional outcomes. Further multicenter trials are required to confirm these findings.

## Introduction

Anatomical endoscopic enucleation of the prostate (AEEP) has revolutionized the surgical treatment of benign prostatic hyperplasia (BPH) by offering a safe, durable, reproducible, and size-independent procedure [1,2]. AEEP is a well-established treatment option that is superior to transurethral resection of the prostate (TURP) in terms of improving prostate symptom scores, enhancing flow rates, and reducing re-treatment rates [3]. AEEP comprises two steps: enucleation of the entire transitional zone of the prostate and tissue removal, the latter of which can significantly affect operative time. Options for tissue evacuation include intravesical mechanical morcellation and the mushroom technique; in selected cases, mini-laparotomy can be performed for cystotomy [4].

Morcellation is currently considered the standard and the most widely used method for AEEP. However, when a mechanical morcellator is unavailable, the mushroom technique is a viable and inexpensive alternative. This method involves preserving a mushroom-like pedicle at the bladder neck, enabling high-speed resection of the avascularized prostate lobes [5]. However, in cases of large prostate glands, this technique can be time-consuming and may increase operative time.

Both the mushroom technique and morcellation are well-established methods for tissue removal and are extensively documented in the current literature. However, evidence of their direct comparative analyses is limited, highlighting a critical knowledge gap on their optimal application. This is especially relevant in relation to prostate size, which our study aims to address. To the best of our knowledge, only one previous publication has compared the efficacy of morcellation with the mushroom technique [6]. The aim of our study was to evaluate the effectiveness of the mushroom technique versus morcellation in en bloc transurethral bipolar prostate enucleation (TUEB) for prostates > 80 mL. We hypothesized that morcellation can decrease operative time and consequently provide a more efficient method for tissue evacuation.

## Materials and methods

We retrospectively reviewed 234 TUEB procedures performed between January 2018 and March 2024 at a single tertiary university hospital. The cohort was divided into the mushroom technique group (n = 116) and the morcellation group (n = 118), based on the acquisition of a morcellator on March 1, 2021, after which the

morcellation technique was systematically used in all procedures. All surgeries were performed by a single surgeon experienced in TUEB who had performed over 500 cases. This study was approved by the Regional and Institutional Ethics Committee, University of Debrecen, Clinical Center (IRB No. DERKEB/IKEB 7116−2025), and written informed consent was obtained from all participants. Data were collected from medical records between April 1, 2025, and April 20, 2025. To ensure anonymity, patient identifiers were removed prior to data analysis, and the authors had no access to any information that could identify individual participants. The inclusion criteria for the study were a prostate volume > 80 mL, maximum flow rate (Qmax) < 15 mL/s, post-void residual (PVR) urine volume > 100 mL, International Prostate Symptom Score (IPSS) > 7, urinary retention, and poor response to medical therapy. Patients who opted out of medical therapy were also included. The exclusion criteria were diagnosis of neurogenic bladder, concurrent bladder stones, urethral stricture, prostate cancer, and any history of TURP or urethral surgery.

Before surgery, all patients underwent a standardized evaluation according to the current European Association of Urology guidelines on Non-Neurogenic Male Lower Urinary Tract Symptoms (EAU Guidelines Edn. presented at the EAU Annual Congress in Madrid in April 2025. ISBN 978-94-92671-29-5).

Demographic data, including age, body mass index (BMI), and American Society of Anesthesiologists (ASA) score, were extracted from clinical records. Prostate specific antigen (PSA) levels were measured, and prostate volume was determined by transabdominal ultrasound (Mindray Diagnostic Ultrasound System, Consona N9, Shenzhen, China) using the ellipsoid volume formula (height × width × length × 0.523). The PSA density of each patient was calculated based on PSA levels and prostate volume. Patients were assessed using the IPSS and Quality of Life (QoL) questionnaires, uroflowmetry, and PVR. Erectile function was evaluated using the International Index of Erectile Function (IIEF-5) questionnaire. Additionally, urine cultures and laboratory parameters, including sodium, creatinine, and hemoglobin levels, were analyzed.

## Surgical technique for en bloc bipolar prostate enucleation using the mushroom technique

Intravenous antibiotics were administered to all patients 30 min before the start of surgery. The procedure was conducted under spinal or general anesthesia with the patient in the lithotomy position. A 27Fr bipolar continuous flow resectoscope (Olympus Winter & IBE GmbH, Hamburg, Germany) equipped with a 12° optic was inserted transurethrally into the bladder using prewarmed normal saline (37°C) as the irrigation fluid. A high frequency bipolar electrode loop (Olympus Winter & IBE GmbH, Hamburg, Germany) was used throughout surgery. A thorough cystoscopy was initially performed to locate the ureteral orifices. The next step involved accessing the prostatic urethra, and an omega-shaped incision was made from the verumontanum to the 12 o'clock position to release the sphincter mucosa. The space between the capsule and adenoma was then opened using the tip of the sheath, starting laterally from the sides of the verumontanum. The left lobe was enucleated by advancing the sheath tip into the bladder at the 2 o'clock position where the vertical fibers of the bladder neck were identified. The adenoma was removed from the 2–6 o'clock position; bleeding on the capsule was coagulated using the loop. For the right lobe, the bladder was entered at the 10 o'clock position, enucleating the adenoma from the top to bottom, and reaching the 6 o'clock region. At this stage, the two enucleation surfaces were connected. A 1–2 cm mushroom-like pedicle was preserved at the 6 o'clock position on the bladder neck to secure the adenoma within the prostatic fossa. The apical portion of the adenoma was enucleated between the 10 o' clock and 2 o'clock positions. The avascularized adenoma was resected using the loop at high speed, and the tissue slices were flushed out (Fig 1A). Final coagulation of the capsule was performed, with particular attention to the bladder neck. The integrity of the sphincter mucosa was verified. A 22Fr three-way urethral catheter was inserted, and continuous bladder irrigation was initiated.

## Surgical technique for en bloc bipolar prostate enucleation using the morcellation technique

The procedure followed the same steps as the mushroom technique, except that no mushroom-like pedicle was preserved at the bladder neck. The enucleated adenoma was then released into the bladder. Once hemostasis was achieved, the outflow was discontinued, and the bladder was filled to increase distension. The instrument was then switched to a

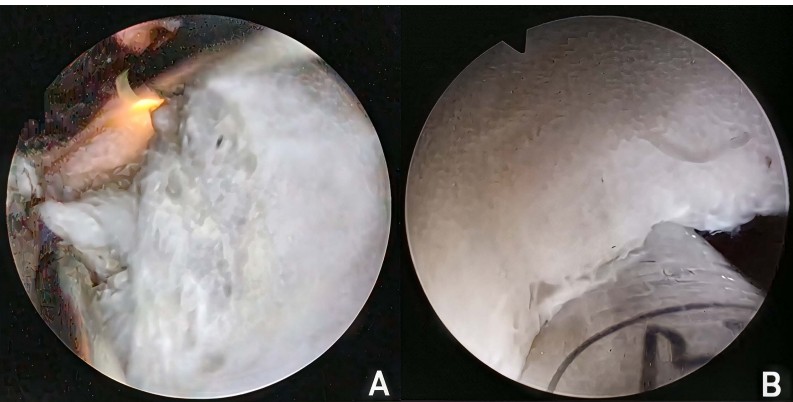

**Fig 1. Intraoperative images of the mushroom (A) and morcellation (B) techniques.**

morcelloscope (Richard Wolf GmbH, Knittlingen, Germany), and an oscillating morcellator (Richard Wolf GmbH, Knittlingen, Germany) with a reusable blade was inserted transurethrally, employing dual irrigation to maintain high intravesical pressure. The morcellator was set to 850 rpm, in accordance with the manufacturer's recommendations (Richard Wolf GmbH). The prostatic adenoma was morcellated in the middle of the bladder by using the conventional upward technique (Fig 1B). A suction pedal was used to capture the adenoma followed by the activation of the morcellation pedal. A 5-L canister was used, which required changing when full. In the presence of smooth, round, hard nodules (referred to as "beach-balls") that bounced off the blade, the morcellation speed was reduced. If the morcellation of these nodules was unfeasible, grasping forceps were used via a morcelloscope for removal. At the end of the procedure, both inflows were discontinued, and the bladder and prostatic fossa were meticulously checked for residual tissue. A 22Fr three-way urethral catheter was inserted, and continuous bladder irrigation was initiated.

## Outcome measures

The operative time starting from the insertion of the bipolar resectoscope until catheter placement was manually recorded. Before formalin fixation, the wet weight of the tissue was measured by a scrub nurse using a weighing scale. The efficacy of the procedures was calculated based on the operative time and the weight of the removed tissue. The occurrence of "beach-balls" was noted in the morcellation cohort. On the first postoperative day, laboratory parameters, including hemoglobin, sodium, and creatinine levels, were monitored, along with the catheterization time and length of hospital stay. The Clavien–Dindo classification was used to assess the 30-day postoperative complications. The outcome measures were compared between the two cohorts.

## Follow-up

Patients with a minimum follow-up duration of 6 months were included in the study. Follow-up was conducted at 1, 3, and 6 months, during which time PSA levels, uroflowmetry, PVR urine, IPSS, QoL, and IIEF-5 scores were measured. Additionally, transient stress urinary incontinence (TSUI) and other late complications were documented.

## Statistical analysis

The Shapiro–Wilk test was used to evaluate the distribution of continuous variables. In cases of non-normal distribution, data are presented as medians with interquartile ranges, and non-parametric methods (Wilcoxon test) were used for comparisons. Categorical variables are presented as counts and percentages. Robust regression analysis was performed

using iteratively reweighted least squares to minimize the influence of outliers on the estimated coefficients. All statistical analyses were conducted in Intercooled Stata version 18.0 (StataCorp LLC, College Station, TX, 2023), with $p < 0.05$ considered statistically significant.

## Results

Patient demographics and preoperative data are presented in Table 1.

The median operative time was significantly longer in the mushroom cohort than in the morcellation cohort, (80 [60–90] min vs. 60 [50–70] min) ($p < 0.001$). The enucleated prostate weight did not differ significantly between the mushroom and morcellation cohorts (68.5 [50–89.5] g vs. 65 [55–89] g) ($p = 0.572$). A significant difference was observed in the median efficacy of the procedures: mushroom at 0.9 [0.76–1.03] g/min and morcellation at 1.25 [1.01–1.44] g/min ($p < 0.001$).

Linear regression analysis showed that prostate volume significantly influenced the operative time of the mushroom technique more than that of the morcellation technique (Graph 1). Each gram of tissue increased the operative time by 0.408 min in the morcellation cohort ($p < 0.05$) and by 0.69 min in the mushroom cohort ($p < 0.05$). Unlike the mushroom technique, the efficacy of morcellation did not decrease with increasing prostate volume. A significant decrease in the median hemoglobin level was observed in both cohorts compared to the preoperative median hemoglobin level, with a

**Table 1. Patient demographics and preoperative data.**

| Variables | Mushroom group | Morcellation group | p value |
|---|---|---|---|
| Patients (n) | 116 | 118 | |
| Age (years), median [IQR] | 70.5 [66–76] | 72 [66–76] | 0.506 |
| BMI (kg/m$^2$), median [IQR] | 28.1 [25–31.3] | 27.73 [24.7–30.7] | 0.249 |
| ASA score 1 [No. (%)] | 4 (3.45%) | 2 (1.69%) | 0.834 |
| ASA score 2 [No. (%)] | 59 (50.86%) | 64 (54.24%) | |
| ASA score 3 [No. (%)] | 52 (44.83%) | 51 (43.22%) | |
| ASA score 4 [No. (%)] | 1 (0.86%) | 1 (0.85%) | |
| Antiplatelet therapy [No. (%)] | 12 (10.34%) | 2 (1.69%) | 0.005 |
| BPH related drugs [No. (%)] | 72 (62.07%) | 58 (49.15%) | < 0.001 |
| Urinary retenion [No. (%)] | 61 (52.59) | 80 (67.80) | 0.017 |
| Prostate volume by ultrasound (ml), median [IQR] | 100 [84–125] | 97.5 [80–120] | 0.353 |
| PSA (ng/mL), median [IQR] | 7.05 [4.13–11.37] | 7.67 [4.14–12] | 0.575 |
| PSA density (ng/mL$^2$), median [IQR] | 0.07 [0.04–0.11] | 0.07 [0.04–0.11] | 0.588 |
| IPSS, median [IQR] | 20 [15.5–24.5] | 20 [14 –23 ] | 0.497 |
| QoL, median [IQR] | 6 [5–6] | 6 [6–6] | 0.027 |
| Qmax (mL/s), median [IQR] | 9.05 [6.45–12.45] | 9 [6.58–12.1] | 0.715 |
| Qave (mL/s), median [IQR] | 4.42 [3–5.83] | 3.87 [2.8–5.3] | 0.204 |
| PVR urine (ml), median [IQR] | 120 [57.5–235] | 100 [50–200] | 0.407 |
| Preoperative urine culture positivity [No. (%)] | 68 (58.62) | 58 (49.15) | 0.146 |
| IIEF–5, median [IQR] | 11 [5 –19 ] | 13 [10 –17 ] | 0.477 |
| Preoperative sodium (mmol/L), median [IQR] | 140 [139–142] | 140 [138–142] | 0.208 |
| Preoperative creatinin (µmol), median [IQR] | 81 [70.5–94] | 85 [75–100] | 0.077 |
| Preoperative Hgb (g/L), median [IQR] | 146.5 [140–153] | 144 [136–152] | 0.123 |

ASA: American Society of Anesthesiologists, BMI: body mass index, BPH: benign prostatic hyperplasia, Hgb: hemoglobin, IIEF–5: International Index of Erectile Function, IPSS: International Prostate Symptom Score, IQR: interquartile range, PSA: prostate specific antigen, PVR: post–void residual, Qave: average flow rate, Qmax: maximum flow rate, QoL: quality of life, TURP: Transurethral Resection of the Prostate.

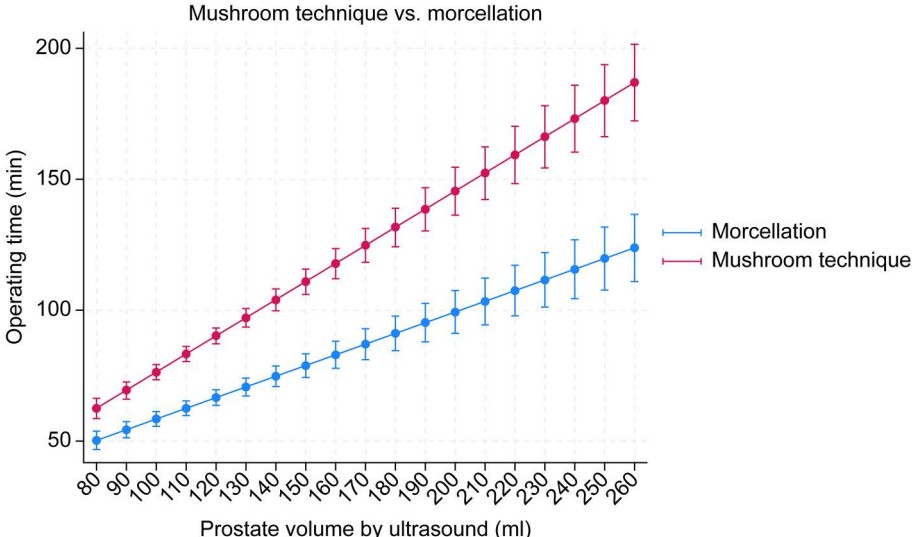

**Graph 1. Linear regression analysis of variation in operative time with prostate size (the entire spectrum of prostate volumes is shown).**

significantly greater drop in the mushroom cohort compared to the morcellation cohort (mushroom: −25 [− 17 to − 8] g/L, morcellation: −11 [− 19 to − 4] g/L, p = 0.002). Since there was a significant difference between the cohorts regarding anti-platelet therapy use and occurrence of urinary retention, we performed a multiple robust regression analysis to account for the potential confounding effects of these factors on operative time and blood loss, as our outcome data were not normally distributed. Unexpectedly, the results revealed that antiplatelet therapy use showed a borderline significant association with a lower hemoglobin decrease (Coef. = −6.53, p = 0.052, 95% CI [−13.11; 0.05]). However, urinary retention did not significantly influence hemoglobin reduction (Coef. = 0.20, p = 0.908, 95% CI [−3.12; 3.51]). Operative time was strongly associated with hemoglobin loss, with each additional minute linked to a greater hemoglobin decrease of 0.13 g/L (Coef. = 0.13, p < 0.001, 95% CI [0.06; 0.20]). The operative technique itself had no significant independent effect on hemoglobin decrease after adjusting for these factors (Coef. = −3.17, p = 0.074, 95% CI [−6.66; 0.31]). Therefore, although operative time significantly impacted blood loss, neither antiplatelet therapy use nor urinary retention occurrence significantly altered the comparative outcomes between the mushroom and morcellation techniques after robust statistical adjustment.

The change between the median preoperative and postoperative sodium levels did not differ significantly between the two cohorts (mushroom: 140 [138–142] mmol/L; morcellation: 140 [138–141] mmol/L, p = 0.889). Similarly, there was no significant difference in median creatinine levels between the two cohorts (mushroom: 79 [67–94] µmol/L, morcellation: 83 [73.5–102] µmol/L, p = 0.902). The median catheterization time and length of hospital stay were significantly shorter in the morcellation cohort (3 [2–3] days vs. 2 [2–3] days, p < 0.001; 4 [3–4] days vs. 3 [3–4] days, p < 0.001, respectively). Conversion to TURP did not occur in either group. Additionally, in the morcellation cohort, there was no need to postpone morcellation to the second stage. The occurrence of "beach-balls" in the morcellation group was 9.32% (n = 11). In nine cases (7.63%), "beach-balls" could be removed with decreased morcellation speed, and in two cases (1.69%), grasping forceps were used to retrieve them from the bladder.

All patients were able to urinate after catheter removal. Histopathological examination revealed BPH in 109 cases (93.97%) and incidental adenocarcinoma in seven cases (6.03%) in the mushroom cohort. In the morcellation cohort, there were 114 cases (96.61%) of BPH and 4 cases (3.39%) of adenocarcinoma. Our analysis showed comparable non-significantly different complication rates between the mushroom and morcellation groups. Regarding Clavien–Dindo grade I complications, gross hematuria was observed in nine patients (7.76%) in the mushroom cohort and in eight patients

(6.78%) in the morcellation cohort. These patients were managed with prolonged bladder irrigation and tranexamic acid. For Clavien–Dindo grade II complications, fever was noted in three patients (2.59%) in the mushroom cohort and in two patients (1.69%) in the morcellation cohort, all of whom were treated with parenteral antibiotic therapy. Clot evacuation and recoagulation (Clavien–Dindo grade IIIb) were needed in two cases (1.69%) in the mushroom cohort and in one case (0.85%) in the morcellation cohort. Blood transfusions were not required in either cohort. No Clavien–Dindo grade IIIa, IV, or V complications occurred in either group. Meanwhile, we observed superficial bladder injury in one patient (0.85%) in the morcellation cohort. Bleeding from the bladder mucosa coagulated, and morcellation continued without complications. Regarding late complications, there was one case each of urethral stricture (0.86%) and bladder neck sclerosis (0.86%) in the mushroom group that was treated with urethrotomy and transurethral bladder neck resection.

At 1 month postoperatively, the incidence of TSUI was 18.10% (n = 21) in the mushroom cohort and 20.33% (n = 24) in the morcellation cohort (p = 0.666). By the 3-month follow-up, TSUI rates decreased to 8.62% (n = 10) in the mushroom group and 10.16% (n = 12) in the morcellation group (p = 0.687), and at 6 months, it further decreased to 4.31% (n = 5) in the mushroom group and 5.08% (n = 6) in the morcellation group (p = 0.781). Regarding the 1-, 3-, and 6-month follow-up, there were no significant differences between the two cohorts in terms of the PSA values, percentage reduction in PSA levels from baseline, Qmax, Qave, PVR, IPSS, QoL, and IIEF-5, except for the Qmax value at the 1-month follow-up (mushroom: 20.85 [15.6–27.4] mL/s vs. morcellation: 17 [11.04–25.45] ml/s, p = 0.022). The detailed follow-up data table is available as supplementary material (S1 Table).

## Discussion

The introduction of AEEP, regardless of the energy source used, marked a paradigm shift in the surgical treatment of BPH over four decades ago, demonstrating its superiority over TURP [7]. A key advantage of AEEP is the achievement of complete de-obstruction, whereas TURP only reduces the prostate volume and results in extra-anatomical outcomes [8]. The second phase of AEEP, which involves tissue removal, is a crucial and time-consuming part of the procedure that significantly affects the overall operative time. Notably, the time required for morcellation accounts for 20%–60% of the total surgical duration [9].

Among the available energy sources, bipolar technology has gained widespread acceptance because of its cost-effectiveness and safety as it delivers less thermal energy and minimizes the risk of TUR syndrome [10,11]. Bipolar prostate enucleation can be performed using either the morcellation or mushroom technique. Neill et al. were the first to publish results on bipolar prostate enucleation using a morcellator, presenting it as a viable alternative to laser enucleation [12]. A morcellator is a complex device with numerous technical challenges that require a surgical team to be thoroughly trained in its appropriate use. Nonetheless, morcellation has been a barrier to the widespread adoption of AEEP owing to the additional equipment and the risk of bladder injury.

Morcellators are categorized according to the blade movement (oscillating vs. reciprocating) and blade shape (toothed vs. non-toothed). Blades can also be either disposable or reusable [13]. Oscillating morcellators are more than twice as efficient as reciprocating ones [14]. Research has compared the effectiveness of different morcellator systems, revealing that *ex vivo* morcellation is significantly faster than *in vivo* procedures [15]. This is because *ex vivo* conditions lack interfering factors such as bleeding, poor visibility, and technical issues that can occur during *in vivo* morcellation. In these studies, homogeneous tissues, such as raw bovine hearts, were used for morcellation. In addition to device speed, several factors can influence the efficacy of *in vivo* morcellation, including the "beach-ball" effect, the weight of the morcellated tissue, PSA density, and the presence of prostate calcification [14]. The occurrence of "beach-ball" is reported in up to 13.4% of cases in the literature, which can reduce the efficiency of morcellation [16]. Herzberg et. al. concluded that "beach-balls" occur more often with older age, larger prostate, indwelling catheter, and chronic urinary retention [17]. Slowing the morcellator speed can help address this problem. If reducing the morcellator speed is ineffective, grasping forceps can be applied through the morcelloscope to remove the hard nodules.

Morcellation of very large prostates can be time-consuming because these glands can fill the entire bladder, and optimal morcellator function requires space at the tip for effective oscillating movement. For extremely large prostates, postponed morcellation may be a viable option [18]. Additionally, a history of urinary tract infections has been associated with an increased morcellation time.

In terms of the learning curve, less experienced surgeons may have slightly longer morcellation times than their more experienced counterparts, but the difference is minimal [19]. One of the main drawbacks of morcellation is the risk of bladder injury, a rare but serious complication. This can occur if the bladder is not adequately distended or if visibility is poor. When this occurs, the surgeon should immediately release the foot pedal to stop suction [9]. To prevent such injuries, morcellation should always be performed after achieving proper hemostasis with good visibility and a well-distended bladder. In our study, we encountered one case of bladder injury. In our case, the injury was a superficial mucosal tear, which was treated with coagulation using the bipolar loop. Morcellation then continued without any further issues. Our bladder injury rate aligns with the findings in the literature [20].

The mushroom technique can be used as an alternative in the absence of a morcellator. Liu et al. reported on 1,100 patients who underwent bipolar prostate enucleation using this technique and concluded that it is a viable alternative to TURP and open prostatectomy [21]. Although this method is less expensive than morcellation, it has some drawbacks. This procedure is time-consuming, and in larger glands, numerous tissue fragments can block the prostate cavity, requiring repeated washouts. Flushing reduces the bladder pressure, which can lead to bleeding from the capsule, impairing visibility and necessitating additional coagulation and blood clot evacuation that further extends the operative time. Surgical time is a critical factor in AEEP procedures as longer operation times can increase the risk of urethral stricture [22].

Although bladder injury is a well-known complication of morcellation, the mushroom technique theoretically poses a risk of bladder explosion, a rare but potentially life-threatening complication. However, this has not been observed in our practice. In the literature, bladder explosion occur primarily with monopolar TURP, although cases involving bipolar techniques have also been documented [23]. Theoretically, lower risk occurs with bipolar methods, since bipolar energy generates less heat, consequently reducing the extent of pyrolysis. This complication arises following the intravesical accumulation of explosive gases formed by the pyrolysis of prostatic tissue and hydrolysis of intracellular water. Ning et al. investigated gas formation during TURP procedures. Hydrogen constitutes a substantial portion of these gases, whereas oxygen is typically present in minimal amounts, insufficient alone to support combustion [24]. For an explosion to occur, atmospheric air containing approximately 21% oxygen must enter the bladder, creating a highly combustible gas mixture. Ignition typically happens when the electrosurgical electrode loop contacts this gas. Risk factors for bladder explosion include prolonged operating time, high-powered cutting and coagulation settings, and inadvertent introduction of the atmospheric air due to leaky connections or improper handling of irrigation systems. Recommended preventive measures include utilizing moderate power settings, ensuring airtight irrigation connections, employing continuous irrigation, and carefully handling the evacuator bulb [25].

To the best of our knowledge, this is the second trial involving a large cohort that compared the effectiveness of the morcellation versus mushroom techniques. Weerasawin et al. demonstrated better functional outcomes and shorter operative time for patients who underwent morcellation compared to those who underwent the mushroom technique [6]. Similarly, in our study, the operative time was significantly shorter in the morcellation cohort. However, the functional results were not influenced by the tissue removal method. This can be attributed to the fact that the fundamental principles of enucleation are similar for both techniques. There was significant blood loss in both cohorts, with greater loss in the mushroom cohort likely due to the longer operative time and the aforementioned reasons. Nonetheless, anemia was clinically insignificant in both groups. The TSUI rate aligns with the rates reported in the literature [26].

Based on our results, the mushroom technique is a suitable alternative for tissue removal during bipolar prostate enucleation. However, for larger glands (> 80 mL), morcellation clearly proves to be a superior option, as its efficiency remains

consistent irrespective of prostate size. This size-dependent difference in efficacy is clinically relevant, as it provides surgeons with practical guidance for tailoring surgical strategy according to prostate volume, ultimately facilitating optimized surgical planning and patient outcomes.

The limitations of our study include its single-center retrospective design, relatively short follow-up period, and lack of randomization, which may affect the generalizability of the data. Another important limitation is that patient inclusion depended on the availability of the morcellator device, which may have introduced selection bias. Although we attempted to mitigate selection bias by applying identical inclusion and exclusion criteria and that all procedures were performed by the same highly experienced surgeon, the division of patients into groups based on morcellator availability rather than specific patient factors may have led to inherent biases. However, all consecutive patients meeting inclusion criteria during the study period were included and the baseline characteristics were comparable between groups, minimizing this risk to some extent. Nevertheless, the retrospective nature of this study and the abovementioned limitations underline the importance of interpreting our results cautiously., The need for future prospective, randomized controlled trials to confirm the generalizability and broader applicability of our findings is highlighted.

## Conclusions

This study demonstrated that morcellation is a more efficient method for tissue removal during bipolar prostate enucleation, particularly in larger prostates, owing to its shorter operative time and consistent efficacy regardless of gland size. Although the mushroom technique remains a viable alternative, it is less efficient for larger prostrates and is associated with longer operative times. Both techniques showed comparable safety profiles and functional outcomes. However, the retrospective design and short follow-up period of this study highlight the need for larger, multicenter trials to validate these findings.

## Supporting information

**S1 Table. Follow-up data.**
(XLSX)

**S2 File. Raw data.**
(XLSX)

## Acknowledgments

We would like to acknowledge Editage's (www.editage.com) support in manuscript preparation. **Patient consent statement:** Written informed consent was obtained from the patients for the publication of this article.

## Author contributions

**Conceptualization:** Zoltán Kiss.

**Data curation:** Zoltán Kiss.

**Formal analysis:** Attila Nagy.

**Methodology:** Zoltán Kiss.

**Supervision:** Tibor Flaskó.

**Writing – original draft:** Zoltán Kiss.

**Writing – review & editing:** Mihály Murányi, Tibor Flaskó.

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
