## [Decision Letter · Decision Letter 0]

3 Jul 2025

PONE-D-25-22862Effectiveness of the Mushroom Technique Versus Morcellation in En Bloc Bipolar Prostate Enucleation for Prostates over 80 mLPLOS ONE

Dear Dr. Zoltán,

Thank you for submitting your manuscript to PLOS ONE. After careful consideration, we feel that it has merit but does not fully meet PLOS ONE’s publication criteria as it currently stands. Therefore, we invite you to submit a revised version of the manuscript that addresses the points raised during the review process.

We look forward to receiving your revised manuscript.

Kind regards,

Bryan Kwun-Chung Cheng

Academic Editor

PLOS ONE

**Journal Requirements:**

1. When submitting your revision, we need you to address these additional requirements. Please ensure that your manuscript meets PLOS ONE's style requirements, including those for file naming. The PLOS ONE style templates can be found at https://journals.plos.org/plosone/s/file?id=wjVg/PLOSOne_formatting_sample_main_body.pdf and https://journals.plos.org/plosone/s/file?id=ba62/PLOSOne_formatting_sample_title_authors_affiliations.pdf 2. In the online submission form, you indicated that “The data that support the findings of this study are available upon reasonable request”.  All PLOS journals now require all data underlying the findings described in their manuscript to be freely available to other researchers, either a. In a public repository, b. Within the manuscript itself, or c. Uploaded as supplementary information.This policy applies to all data except where public deposition would breach compliance with the protocol approved by your research ethics board. If your data cannot be made publicly available for ethical or legal reasons (e.g., public availability would compromise patient privacy), please explain your reasons on resubmission and your exemption request will be escalated for approval. 3. When completing the data availability statement of the submission form, you indicated that you will make your data available on acceptance. We strongly recommend all authors decide on a data sharing plan before acceptance, as the process can be lengthy and hold up publication timelines. Please note that, though access restrictions are acceptable now, your entire data will need to be made freely accessible if your manuscript is accepted for publication. This policy applies to all data except where public deposition would breach compliance with the protocol approved by your research ethics board. If you are unable to adhere to our open data policy, please kindly revise your statement to explain your reasoning and we will seek the editor's input on an exemption. Please be assured that, once you have provided your new statement, the assessment of your exemption will not hold up the peer review process. 4. Your ethics statement should only appear in the Methods section of your manuscript. If your ethics statement is written in any section besides the Methods, please delete it from any other section.

Reviewers' comments:

Reviewer's Responses to Questions

**Comments to the Author**

1. Is the manuscript technically sound, and do the data support the conclusions?

Reviewer #1: Partly

Reviewer #2: Yes

2. Has the statistical analysis been performed appropriately and rigorously? 

Reviewer #1: N/A

Reviewer #2: Yes

3. Have the authors made all data underlying the findings in their manuscript fully available?

Reviewer #1: Yes

Reviewer #2: Yes

4. Is the manuscript presented in an intelligible fashion and written in standard English?

Reviewer #1: Yes

Reviewer #2: Yes

5. Review Comments to the Author

**Reviewer #1:**  This study compared the mushroom technique with morcellation for en bloc transurethral bipolar prostate enucleation (TUEB) in prostate. Morcellation significantly reduced operative time and improved tissue removal efficiency, with both techniques showing similar safety. The authors conclude that morcellation is preferable for larger prostates, though both are viable options. We have the concerns as the followings:

1. Considering that both the mushroom technique and morcellation have been widely used in clinical practice for years, could the authors clarify what the key innovation or new contribution of this study is compared to the existing literature?

2. What were the specific types and rates of complications in each group, beyond the need for transfusion or conversion to TURP? For instance, were there differences in urinary incontinence, urethral strictures, or bladder injuries?

3. Bladder injury is one of the main complications of morcellation. Was bladder explosion observed during the mushroom technique? Please elaborate on this point in the discussion section.

4. Were there any criteria for selecting between the mushroom technique and morcellation in individual patients, or was allocation purely based on surgeon preference?

5. Please reorganize the tables to make more concise and clear.

**Reviewer #2:**  This study provides valuable insights into the efficacy and safety of the mushroom technique compared to morcellation in en bloc bipolar prostate enucleation for large prostates. The manuscript is well-structured and addresses an understudied aspect of anatomical endoscopic enucleation of the prostate. However, certain methodological and analytical limitations should be addressed to strengthen the conclusions:

1. Given the retrospective nature of the study, the lack of randomization and the division of cohorts based on equipment availability rather than patient characteristics may introduce potential selection bias. A discussion of how this may affect generalizability is warranted.

2. Table 1 reveals significant differences between groups in antiplatelet therapy use and urinary retention rates. Since these factors may influence operative time and blood loss, they should be accounted for in comparative analysis.

3. The reported prostate volume in Graph 1 (up to 260 mL) conflicts with the ultrasound-derived maximum volume of 125 mL in Table 1. Clarification is needed on whether specimen weights (post-enucleation) were inadvertently compared to preoperative ultrasound measurements, and any outliers should be addressed.

4. To improve clarity, consider removing non-significant outcomes from Table 2 or consolidating them into supplementary material.

6. PLOS authors have the option to publish the peer review history of their article (what does this mean? ). If published, this will include your full peer review and any attached files.

**Do you want your identity to be public for this peer review?** For information about this choice, including consent withdrawal, please see our Privacy Policy .

Reviewer #1: No

Reviewer #2: No

---

## [Author Response · Author response to Decision Letter 1]

23 Jul 2025

Reviewer #1:

This study compared the mushroom technique with morcellation for en bloc transurethral bipolar prostate enucleation (TUEB) in prostate. Morcellation significantly reduced operative time and improved tissue removal efficiency, with both techniques showing similar safety. The authors conclude that morcellation is preferable for larger prostates, though both are viable options. We have the concerns as the followings:

1. Considering that both the mushroom technique and morcellation have been widely used in clinical practice for years, could the authors clarify what the key innovation or new contribution of this study is compared to the existing literature?

Thank you for raising this valuable point. We fully agree that both the mushroom technique and morcellation are well-established methods for tissue removal, and are extensively documented in the current literature. Indeed, several studies have previously explored the effectiveness of various morcellators and morcellation strategies (including upwards and downwards techniques). Nonetheless, we identified only one prior study that directly compared the mushroom technique against morcellation.

Our present study substantially expands upon this previous work, notably by including more than twice the number of patients, which significantly strengthens the reliability and applicability of our results. We believe that our findings provide valuable insight into a critical limitation of the mushroom technique compared to morcellation, that is, as prostate volume increases, the efficiency of the mushroom technique decreases significantly, whereas morcellation consistently maintains its effectiveness. This highlights a clear clinical implication that surgeons intending to employ the mushroom technique should carefully consider this limitation, particularly when managing larger prostates (above 80 mL), to ensure optimal surgical outcomes and operative efficiency. Based on your question, we have expanded the introduction and discussion sections of the manuscript accordingly.

Lines: 60-64, 352-357

2. What were the specific types and rates of complications in each group, beyond the need for transfusion or conversion to TURP? For instance, were there differences in urinary incontinence, urethral strictures, or bladder injuries?

Thank you for your insightful question regarding the complications. Our analysis showed comparable complication rates between the mushroom and morcellation groups.

Regarding Clavien–Dindo grade I complications, gross hematuria was observed in nine patients (7.76%) in the mushroom cohort and in eight patients (6.78%) in the morcellation cohort. These patients were managed with prolonged bladder irrigation and tranexamic acid. For Clavien–Dindo grade II complications, fever was noted in three patients (2.59%) in the mushroom cohort and in two patients (1.69%) in the morcellation cohort, all of whom were treated with parenteral antibiotic therapy. Clot evacuation and recoagulation (Clavien–Dindo grade IIIb) were needed in two cases (1.69%) in the mushroom cohort and in one case (0.85%) in the morcellation cohort. Blood transfusions were not required in either cohort. No Clavien–Dindo grade IIIa, IV, or V complications occurred in either group. Meanwhile, we observed superficial bladder injury in one patient (0.85%) in the morcellation cohort. Bleeding from the bladder mucosa coagulated, and morcellation continued without complications. Regarding late complications, there was one case each of urethral stricture (0.86%) and bladder neck sclerosis (0.86%) in the mushroom group that was treated with urethrotomy and transurethral bladder neck resection.

At 1 month postoperatively, the incidence of TSUI was 18.10% (n=21) in the mushroom cohort and 20.33% (n=24) in the morcellation cohort (p=0.666). By the 3-month follow-up, TSUI rates decreased to 8.62% (n=10) in the mushroom group and 10.16% (n=12) in the morcellation group (p=0.687), and at 6 months, it further decreased to 4.31% (n=5) in the mushroom group and 5.08% (n=6) in the morcellation group (p=0.781).

We have expanded the results section of the manuscript accordingly.

Lines: 238-257

3. Bladder injury is one of the main complications of morcellation. Was bladder explosion observed during the mushroom technique? Please elaborate on this point in the discussion section.

Thank you for this important question. Indeed, bladder injury is a known complication of morcellation and most of these injuries are superficial mucosal lesions, whereas bladder perforation is relatively rare. Fortunately, we did not observe bladder explosion in our practice; however, following your suggestion, we have elaborated in detail on the underlying mechanisms of this complication in the discussion section, and supplemented the text with relevant additional references.

Lines: 323-340

Reference numbers: 23-25

4. Were there any criteria for selecting between the mushroom technique and morcellation in individual patients, or was allocation purely based on surgeon preference?

Thank you for highlighting this important methodological point. We did not apply specific selection criteria for choosing between the mushroom technique and morcellation, which represents a limitation of our study. Since March 1, 2021, after acquiring a morcellator at our institution, we have consistently employed morcellation in cases involving prostates over 80 mL to shorten the operative time and improve surgical efficiency. We acknowledge that this approach may have introduced selection bias; therefore, we explicitly addressed this limitation in the Discussion section of our manuscript. Future randomized controlled studies may help clarify the comparative effectiveness of these methods further, minimizing potential biases.

Lines: 360-370

5. Please reorganize the tables to make more concise and clear.

Thank you for this suggestion. The reorganization was made accordingly, resulting in improved clarity and conciseness.

Reviewer #2: This study provides valuable insights into the efficacy and safety of the mushroom technique compared to morcellation in en bloc bipolar prostate enucleation for large prostates. The manuscript is well-structured and addresses an understudied aspect of anatomical endoscopic enucleation of the prostate. However, certain methodological and analytical limitations should be addressed to strengthen the conclusions:

1. Given the retrospective nature of the study, the lack of randomization and the division of cohorts based on equipment availability rather than patient characteristics may introduce potential selection bias. A discussion of how this may affect generalizability is warranted.

We agree with the reviewer's comments regarding the potential selection bias arising from the retrospective design of our study, the lack of randomization, and the division of patient cohorts based primarily on equipment availability (i.e., introduction of morcellation after acquiring the morcellator) rather than patient-specific characteristics.

We have expanded the discussion section to address these limitations accordingly.

Lines: 360-370

2. Table 1 reveals significant differences between groups in antiplatelet therapy use and urinary retention rates. Since these factors may influence operative time and blood loss, they should be accounted for in comparative analysis.

Thank you for your valuable comment. We performed a multiple robust regression analysis to account for the potential confounding effects of antiplatelet therapy use and urinary retention on operative time and blood loss, as our outcome data were not normally distributed.

We have expanded the results section accordingly.

Lines: 173-175, 204-218

3. The reported prostate volume in Graph 1 (up to 260 mL) conflicts with the ultrasound-derived maximum volume of 125 mL in Table 1. Clarification is needed on whether specimen weights (post-enucleation) were inadvertently compared to preoperative ultrasound measurements, and any outliers should be addressed.

Thank you very much for highlighting this important point. The prostate volumes reported in Table 1 are median values with interquartile ranges (IQR), due to the non-normal distribution of the data from our study population. Specifically, the 125 mL mentioned in Table 1 reflects the upper quartile (75th percentile) of prostate volume, not the maximum measured value. In fact, the largest prostate volume measured preoperatively by ultrasound in our cohort was 260 mL, consistent with the maximum value illustrated in Graph 1. To clarify this, we have explicitly stated as follows: Graph 1 reflects the full range of prostate volumes measured, including the maximum observed value.

Lines: 232-233

4. To improve clarity, consider removing non-significant outcomes from Table 2 or consolidating them into supplementary material.

Thank you very much for your suggestion. Considering that we observed statistically significant difference only in the maximum flow rate (Qmax) at 1 month, we have decided to highlight this result explicitly within the Results section of our manuscript. Consequently, we have removed the entire Table 2 from the main document and included it as a supplementary material, to improve clarity of the manuscript (Table S1).

Lines: 257-262

---

## [Decision Letter · Decision Letter 1]

12 Aug 2025

Effectiveness of the Mushroom Technique Versus Morcellation in En Bloc Bipolar Prostate Enucleation for Prostates over 80 mL

PONE-D-25-22862R1

Dear Dr.Kiss Zoltán,

We’re pleased to inform you that your manuscript has been judged scientifically suitable for publication and will be formally accepted for publication once it meets all outstanding technical requirements.

Kind regards,

Bryan Kwun-Chung Cheng

Academic Editor

PLOS ONE

Additional Editor Comments (optional):

Reviewers' comments:

Reviewer's Responses to Questions

**Comments to the Author**

1. If the authors have adequately addressed your comments raised in a previous round of review and you feel that this manuscript is now acceptable for publication, you may indicate that here to bypass the “Comments to the Author” section, enter your conflict of interest statement in the “Confidential to Editor” section, and submit your "Accept" recommendation.

Reviewer #2: All comments have been addressed

2. Is the manuscript technically sound, and do the data support the conclusions?

Reviewer #2: Yes

3. Has the statistical analysis been performed appropriately and rigorously? 

Reviewer #2: Yes

4. Have the authors made all data underlying the findings in their manuscript fully available?

Reviewer #2: Yes

5. Is the manuscript presented in an intelligible fashion and written in standard English?

Reviewer #2: Yes

6. Review Comments to the Author

Reviewer #2: (No Response)

7. PLOS authors have the option to publish the peer review history of their article (what does this mean? ). If published, this will include your full peer review and any attached files.

**Do you want your identity to be public for this peer review?** For information about this choice, including consent withdrawal, please see our Privacy Policy .

Reviewer #2: No

---

## [Editor Report · Acceptance letter]

PONE-D-25-22862R1

PLOS ONE

Dear Dr. Zoltán,

I'm pleased to inform you that your manuscript has been deemed suitable for publication in PLOS ONE. Congratulations! Your manuscript is now being handed over to our production team.

Kind regards,

on behalf of

Dr. Bryan Kwun-Chung Cheng

Academic Editor

PLOS ONE